# The Impact of Alcohol on Sleep Physiology: A Prospective Observational Study on Nocturnal Resting Heart Rate Using Smartwatch Technology

**DOI:** 10.3390/nu17091470

**Published:** 2025-04-26

**Authors:** Anna Strüven, Jenny Schlichtiger, John Michael Hoppe, Isabel Thiessen, Stefan Brunner, Christopher Stremmel

**Affiliations:** 1Department of Medicine I, LMU University Hospital, LMU Munich, 81377 Munich, Germany; 2DZHK (German Centre for Cardiovascular Research), Partner Site Munich Heart Alliance, LMU University Hospital, LMU Munich, 81377 Munich, Germany; 3Department of Medicine IV, LMU University Hospital, LMU Munich, 80336 Munich, Germany

**Keywords:** alcohol, sleep, resting heart rate, smartwatch

## Abstract

**Background/Objectives:** Alcohol consumption is known to influence cardiovascular regulation and sleep quality; however, real-world data on its acute effects—particularly during nocturnal rest—are limited. This study aimed to investigate the impact of moderate alcohol intake on nocturnal resting heart rate (HR) and sleep parameters using continuous smartwatch-based monitoring in healthy individuals. **Methods:** In this prospective observational study, 40 healthy adults (63% female, mean age of 30.5 years) underwent a structured 9-day smartwatch monitoring period. The protocol included three alcohol-free baseline days, three consecutive evenings with moderate alcohol consumption (40 g/day for women, 60 g/day for men), and three post-exposure days. Continuous data on HR, sleep stages, nocturnal awakenings, and physical activity were recorded. Subjective sleep quality was assessed using the Pittsburgh Sleep Quality Index (PSQI) at baseline. The primary endpoint was the change in the average nocturnal resting HR. Secondary outcomes included sleep parameters and activity levels. **Results:** Alcohol consumption led to a statistically significant increase in nocturnal resting HR from 63.6 ± 9.2 bpm at baseline to 66.6 ± 9.0 bpm during exposure (*p* < 0.001), with rapid normalization during the post-exposure phase (64.9 ± 9.3 bpm). No significant changes were observed in objective sleep architecture or daytime activity. Despite stable sleep structure, participants reported reduced subjective sleep quality under alcohol exposure, suggesting a potential link to the elevated HR. **Conclusions:** Even moderate alcohol intake transiently elevates nocturnal resting HR without affecting sleep architecture, likely impairing physiological recovery. These findings underscore the underestimated cardiovascular impact of alcohol and warrant further research in larger and more diverse populations.

## 1. Introduction

The effects of alcohol consumption on cardiovascular and sleep physiology have long been a subject of scientific investigation [1]. Traditionally, moderate alcohol intake—defined as one to two standard drinks per week (i.e., 14 to 28 g of pure alcohol)—was associated with potential cardioprotective effects, particularly through the consumption of polyphenol-rich red wine [2,3,4]. However, subsequent studies suggested that these observed benefits were likely confounded by generally healthier lifestyle factors rather than attributable to alcohol itself [4].

Today, large-scale meta-analyses demonstrate dose-dependent harmful effects of alcohol, independent of beverage type. Even minimal alcohol exposure has been associated with an increased risk of alcohol-related cancers, and the adjusted relative risk for all-cause mortality is 1.05 for medium-volume drinkers (25–45 g/day) and 1.19 for high-volume drinkers (45–65 g/day), with disproportionately higher risks observed in women [5,6]. Consequently, the World Health Organization (WHO) now states that no level of alcohol consumption can be considered safe for health [7].

The impact of alcohol on nocturnal physiological recovery processes, including sleep quality and cardiovascular regulation, remains insufficiently understood in real-world contexts. Sleep quality plays a pivotal role in overall health, influencing cognitive performance, metabolic regulation, and cardiovascular risk. Nighttime is typically characterized by a predominance of parasympathetic nervous system activity, allowing the body to recover. Disruptions to this autonomic balance—such as an elevated nocturnal resting heart rate (HR)—are associated with reduced sleep quality and an increased long-term risk of hypertension, atrial fibrillation, and cardiovascular mortality [1,6,8,9].

Alcohol is a well-established modulator of the autonomic nervous system. Acute intake leads to increased sympathetic activity and reduced vagal tone, resulting in elevated HR and decreased HR variability [10,11,12,13]. Studies have shown that even small amounts of alcohol can trigger measurable increases in resting HR within 1 to 3 h of consumption [1,14,15]. The MunichBREW study, for example, demonstrated a linear relationship between breath alcohol concentration and HR during real-world exposure [8]. Moreover, controlled trials confirm that this elevation can persist for several hours into the night, with effects that are dose-dependent and primarily driven by ethanol itself, regardless of beverage type [11,15,16].

Despite these findings, data on alcohol’s effects on cardiovascular physiology during sleep remain limited—particularly in real-world settings with the use of continuous, wearable-based monitoring. Existing research often relies on laboratory conditions or commercial applications with limited scientific validation [11,16,17]. Furthermore, while alcohol is widely perceived to impair sleep, objective analyses of sleep architecture frequently reveal minimal changes in parameters such as total sleep time or sleep phase distribution. A large meta-analysis showed that high alcohol doses (>0.85 g/kg) can reduce sleep onset latency, but this effect is absent with lower doses. Total sleep duration remains largely unchanged regardless of alcohol intake; however, even moderate doses (1–2 drinks/day) can reduce rapid eye movement (REM) sleep by approximately 10–15 min, while light and deep sleep stages are largely unaffected [18,19]. Subjective sleep quality assessments vary widely across studies, raising the question of whether the perceived decline in sleep quality may be linked more to physiological dysregulation (e.g., elevated HR) than to altered sleep architecture.

In this prospective observational study, we investigated the acute effects of moderate alcohol consumption on nocturnal resting HR using continuous smartwatch-based monitoring. We further examined potential associations with objective sleep architecture, daily physical activity, and subjective sleep quality. By combining subjective and objective data in a naturalistic setting, this study aims to provide new insights into the subtle but clinically relevant interactions between alcohol intake, sleep physiology, and cardiovascular function. By leveraging wearable-based monitoring in a real-world context, this study offers novel insights into interactions that are often missed in controlled laboratory environments.

## 2. Materials and Methods

### 2.1. Study Population

This prospective observational cohort study was conducted at the Ludwig Maximilians University (LMU) Hospital in Munich, Germany, between 7 July 2024 and 15 January 2025. A total of 40 healthy subjects were included in this study. The exclusion criteria were age under 18 years, any chronic illness, a general refusal to consume alcohol as described in the study protocol, and a known addiction or other psychiatric illness.

The study was conducted in accordance with the tenets of the Declaration of Helsinki and the German Data Protection Act. All subjects provided informed consent prior to inclusion. The study protocol was reviewed and approved by the institutional ethics committee of LMU Munich, Germany (#23-0972).

### 2.2. Data Acquisition and Study Protocol

Each subject was provided with a Withings Scanwatch (Withings, Issy-les-Moulineaux, France) to continuously track activity, HR, and sleep parameters (duration, sleep phases, duration and number of awake phases, sleep latency, wake-up duration). Charging was not required during the 9-day study period (3 days of pre-exposure baseline, 3 days of exposure to alcohol in the evening, 3 days of post-exposure).

The amount of alcohol consumed during the exposure period was 40 g per day for women and 60 g per day for men. The type of alcoholic beverage (e.g., wine or beer) was left up to the test subjects. Participants were instructed not to consume any alcohol during pre- and post-exposure measurements and to drink an equal amount of water on these days to exclude volume-mediated effects. Optimum adherence to the study protocol was controlled and achieved by daily written reports on consumed beverages (type, volume, time) and physical activity.

In addition to our smartwatch recordings, participants were asked to complete specific questionnaires before and after the investigation period. To assess subjective baseline sleep quality, we used the Pittsburgh Sleep Quality Index (PSQI) with its standardized questionnaire as well as questions on general sleep quality, the duration taken to fall asleep, nightly awakenings, and restful sleep for baseline versus alcohol exposure comparisons.

### 2.3. Outcome Measures

The primary endpoint in this observational study was alcohol-related differences in average nighttime HR. Secondary endpoints were exposure-related changes in other sleep-associated parameters, differences in daily activity, and the evaluation of subjective parameters assessed through questionnaires.

### 2.4. Statistical Analysis

Categorical variables are expressed as percentages. Normality was confirmed with the Shapiro–Wilk test, with continuous variables being presented as means and standard deviations (SDs). Comparisons between groups were conducted for the primary endpoint of average HR using a two-sided paired *t*-test. Secondary endpoints were evaluated using one-way ANOVA with Geisser–Greenhouse correction and Sidak’s multiple comparisons test. For contingency analyses of subjective sleep reports, we used Fisher’s exact test. All statistical analyses were performed using GraphPad Prism 10 (GraphPad Software, LLC, San Diego, CA, USA).

Due to the exploratory design of the study, no prior sample size calculation was performed. To evaluate the statistical sensitivity of the test, a post hoc power analysis was performed in G*Power (Version 3.1.9.6) using the observed effect size (Cohen’s *d* = 0.58; mean difference = 2.277; SD of differences = 3.948). The analysis indicated a statistical power of 94% (α = 0.05, two-tailed), suggesting adequate sensitivity to detect a medium-sized effect.

## 3. Results

### 3.1. Baseline Characteristics

A total of 40 healthy participants were included in this prospective observational study. The cohort was predominantly female (63%, N = 25/40), with a mean age of 30.5 years and a mean body mass index (BMI) of 25.2 kg/m^2^, and 25% (N = 10/40) of participants were current or former smokers. Regular alcohol consumption, defined as intake on at least one day per week, irrespective of quantity, was reported by half of all participants (N = 20/40). However, the self-reported average daily alcohol consumption was low at 9.4 ± 7.6 g. Regarding physical activity, 65% of the cohort reported engaging in regular exercise (N = 27/40). The self-estimated average daily step count was approximately 8500 steps.

There was an almost equal distribution during the exposure phase in terms of the predominant source of alcohol, with 21 (53%) participants drinking beer (43% female) and 19 (48%) participants drinking wine (84% female). This translated to an alcohol-related calorie intake of 467 ± 76 in the beer group vs. 258 ± 89 in the wine group.

### 3.2. Subjective Sleep Quality

Subjective sleep quality was assessed at baseline using the Pittsburgh Sleep Quality Index (PSQI). The mean PSQI score was 6.2 ± 2.5. Based on established thresholds, 45% of participants were categorized as having good sleep quality, 50% were categorized as moderate, and 5% met the criteria for chronic sleep disturbances. Further analysis of individual PSQI components indicated that prolonged sleep latency and daytime sleepiness were the main contributors to diminished overall sleep quality at baseline (Table 1). Interestingly, alcohol exposure led to a subjective decrease in general sleep quality in almost half (45%, N = 18/40) of all participants, while it was unchanged in 50% (N = 20/40) and improved in 5% (N = 2/40) (Figure 1). When comparing subjective baseline assessments with those after alcohol exposure, we observed a significant increase in nightly awakenings (*p* < 0.001) and a decrease in perceived restful sleep (*p* = 0.017). Yet difficulties in falling asleep were unaffected by moderate alcohol exposure in our study (*p* = 0.922) (Figure 1).

### 3.3. Objective Sleep Parameters Assessed by Smartwatch Monitoring

Objective sleep data obtained through smartwatch-based monitoring showed strong concordance with subjective reports. No significant changes were observed in core sleep parameters in relation to alcohol exposure. The mean duration of light sleep was 4.52 ± 1.13 h, while deep sleep averaged 3.47 ± 0.88 h, with no significant variation across study phases (pre vs. alcohol: light sleep 95% CI −0.04 to +0.38 h, *p* = 0.339; deep sleep 95% CI −0.17 to +0.47 h, *p* = 0.116). Participants were awake for an average of 23 ± 12 min per night and experienced approximately 1.7 ± 1.1 nocturnal awakenings. Mean sleep latency ranged from 3 to 6 min, indicating efficient sleep initiation regardless of alcohol consumption. Similarly, the time from final awakening to getting out of bed averaged 4 min across all study phases (Table 2).

### 3.4. Physical Activity

Continuous monitoring enabled the uninterrupted tracking of physical activity throughout the study period. The mean basal metabolic rate was approximately 1500 kcal/day, with an additional mean activity-related energy expenditure of 300 kcal/day. Participants covered an average distance of 6000 m per day, corresponding to approximately 8000 steps. No significant differences in physical activity parameters were observed across the pre-exposure, exposure, and post-exposure phases (Table 2).

### 3.5. Resting Heart Rate

The primary cardiovascular outcome was nocturnal resting HR. Prior to alcohol exposure, the mean resting HR was 63.6 ± 9.2 bpm. During the alcohol exposure phase, a statistically significant increase of 3.0 bpm was observed to 66.6 ± 9.0 bpm (95% CI −4.77 to −1.37, *p* < 0.001) (Table 2, Figure 1). Following the cessation of alcohol intake, resting HR returned rapidly to near-baseline levels (64.9 ± 9.3 bpm). Minimum and maximum HR values did not differ significantly between phases, remaining stable at approximately 55 bpm and 85 bpm, respectively (Table 2).

Taking a closer look at different subgroups for the primary endpoint of resting HR, we found more pronounced effects in women vs. men and people with a BMI < 25 kg/m^2^ vs. those with a BMI > 25 kg/m^2^, as well as in current or former smokers vs. non-smokers (Appendix A). Interestingly, reported regular physical activity or regular low-dose alcohol consumption (<10 g/day) did not have an influence in our subgroup quantifications (Appendix A).

## 4. Discussion

This prospective observational study aimed to investigate the acute effects of low-to-moderate alcohol consumption on nocturnal resting HR and sleep physiology using continuous, smartwatch-based monitoring under real-world conditions. The principal finding was a statistically significant increase in nocturnal resting HR following alcohol intake, while objective sleep parameters remained largely unaffected.

Even low-to-moderate alcohol consumption resulted in a measurable increase of approximately 3 bpm in the average nocturnal HR. Interestingly, our subgroup analysis indicates more pronounced effects in women with a BMI < 25 kg/m^2^. Possible reasons include gender-specific autonomic responses and higher effective blood alcohol concentrations, which were not assessed in our study. Further research is necessary to validate these exploratory observations.

The seemingly small elevation in HR upon moderate alcohol exposure may reflect a relevant autonomic imbalance, likely driven by increased sympathetic activity and/or reduced parasympathetic tone during sleep. The suppression of the vagal system is also reflected by reduced HR variability under alcohol exposure in previous investigations [13,20]. Potentially involved mechanisms include the vasodilatory effect of alcohol, leading to baroreceptor-mediated sympathetic activation as well as direct chronotropic effects of the activation of calcium channels, and increased plasma levels of adrenalin and cortisol [12,21]. The above-mentioned vasodilatory effect can also lead to heat loss with a subsequent thermoregulatory stress reaction. Other stress responses might be inflammatory in nature, potentially due to the toxic properties of alcohol itself or an electrolyte imbalance because of alcohol-mediated dehydration [12,21].

A long-term increase in resting HR has been associated with increased cardiovascular risk and mortality in several studies [22]. Although a 3 bpm increase may appear modest at first glance, it may nevertheless reflect a clinically relevant autonomic imbalance. Previous analyses have shown that each 10 bpm increase in nighttime HR is associated with a 10% increase in all-cause and cardiovascular mortality. Of note, this association is linear and shows consistent results in several studies, allowing us to calculate a predicted increase in mortality of about 3% based on the 3 bpm HR increase in our study [22,23].

Our findings are consistent with the existing literature describing a dose-dependent relationship between alcohol and HR elevation by about 5 bpm under moderate alcohol consumption, potentially contributing to cardiovascular strain even in healthy individuals [1,15,16]. In laboratory conditions, de Zambotti et al. reported a 4% increase in HR under low-dose alcohol exposure, which increased up to 14% under high doses [20]. Notably, the observed changes occurred acutely and reversed rapidly following alcohol cessation, highlighting their transient nature in this context. Short atrial runs or episodes of atrial fibrillation might be partially responsible for an increased nocturnal HR [8,9]. Yet these episodes were not recorded by our smartwatches during the study period, and we did not observe a significant impact on maximum HR in our data.

There is only a very limited number of studies with small cohorts investigating alcohol-mediated effects during the night. These studies were either limited by their artificial nature in a sleep laboratory setting or did not record sleep parameters at all [11,13,16,17]. Therefore, we based our approach on smartwatch technology to overcome these limitations.

Interestingly, no significant changes were detected in objective sleep architecture—neither in total sleep time, the proportions of light and deep sleep, and the number of awakenings, nor in sleep latency. This is in line with data from a large meta-analysis on objective sleep parameters upon moderate alcohol exposure, which identified an approximately 10 min reduction in REM sleep duration as the only significant alteration [18]. However, smartwatch-based sleep surveillance is largely not capable of tracking REM sleep phases. Despite stable objective sleep parameters, participants reported a decline in perceived sleep quality during alcohol exposure. Specifically, self-reported nighttime awakenings increased, while the sensation of restorative sleep declined. These effects were not mirrored by changes in sleep architecture, suggesting that elevated nocturnal HR may impair physiological recovery and thereby influence subjective sleep experience. Our data suggest that even a modest increase in HR may disrupt the body’s ability to achieve a restorative state during sleep, thereby impairing perceived sleep quality. This finding is in line with a previous study, which demonstrated a clear association between increased nighttime HR and reduced sleep quality measured by the PSQI [24]. Importantly, daily physical activity remained stable across all study phases, ruling out changes in activity level as a confounding factor. Moreover, alcohol exposure itself did not significantly alter daytime behavior, suggesting that the observed alterations in nocturnal HR were not mediated by variations in energy expenditure or physical exertion.

Several limitations should be acknowledged. The relatively small sample size may have limited our ability to detect more subtle changes in physiological parameters beyond HR. Although a post hoc power analysis confirmed sufficient sensitivity to detect the observed HR changes (power: 94%, Cohen’s *d* = 0.58), larger cohorts are needed to validate and expand on these findings. Moreover, the observation period was limited as the study focused on short-term effects of moderate alcohol consumption. Effects of chronic consumption or long-term effects are beyond the scope of this investigation.

In line with most previous studies, we did not restrict alcohol exposure to a specific beverage (e.g., wine or beer). While this enhances the ecological validity and generalizability of our results, it reduces experimental control over potentially relevant confounders. It is conceivable that beverage-specific components—such as sugar, polyphenols, or caloric content—may influence nocturnal HR or sleep physiology, but our study was not designed to isolate these effects. However, a subgroup analysis differentiating between the type of consumed beverage with its respective energy content did not reveal any clear differences between these two cohorts. Recent data on the widely debated cardioprotective effects of red wine suggest that the reported benefits may be more attributable to confounding by a healthier lifestyle than to wine itself [4,5,6]. In line with this claim, the estimated required amount of wine to achieve protective polyphenol blood levels is far beyond the doses investigated in this study [25].

Our study population consisted exclusively of healthy adults. Therefore, the findings cannot be extrapolated to individuals with pre-existing medical conditions or those taking chronic medications, and future research should address these specific subgroups. Another potential limitation lies in the psychological context of alcohol consumption. Participants were aware of their alcohol intake, which may have introduced expectation-driven bias. The perceived decline in sleep quality could in part be attributed to a placebo or nocebo effect, reflecting an anticipated deterioration in sleep following alcohol exposure rather than a true physiological change. As volume overload might have influenced our results, we asked our participants to consume an equal amount of water on control days and to document this in a daily report.

In terms of measurement accuracy, the use of commercial smartwatch technology represents both a strength and a constraint. While wearables allow for continuous, real-world data collection, they are limited in their ability to detect certain sleep stages, particularly REM sleep, which was not recorded by the device used in this study. As a result, subtle changes in sleep architecture—especially those affecting REM sleep—may have gone undetected due to technological constraints rather than a true absence of effect. The scientific validation of commercial devices such as the Withings Scanwatch remains limited, and previous research suggests that their performance, particularly in sleep staging, lags behind that of polysomnography and research-grade wearables [26,27,28]. Nevertheless, we believe that commercial wearable technologies will continue to improve and may play a pivotal role in enabling large-scale, real-world investigations in sleep and cardiovascular research.

## 5. Conclusions

This study demonstrates that even low-to-moderate alcohol consumption leads to a transient yet statistically significant increase in nocturnal resting HR in healthy individuals. While objective sleep architecture remained unchanged, the observed elevation in HR may have impaired physiological recovery during sleep and may potentially explain the frequently reported decline in subjective sleep quality following alcohol exposure.

These effects were independent of daytime physical activity and resolved rapidly after the cessation of alcohol intake, indicating an acute and reversible autonomic response. Although long-term outcomes were beyond the scope of this study, prior research has linked chronically elevated resting HR to an increased risk of cardiovascular morbidity and mortality.

Our findings underscore that alcohol intake—even at levels commonly perceived as harmless—can have immediate effects on cardiovascular regulation during sleep. From a public health perspective, these results support the recommendation to avoid alcohol consumption, particularly for individuals with cardiovascular risk factors or sleep disturbances. Further smartwatch-based studies involving larger, more diverse populations and extended observation periods are warranted to explore the chronic effects of alcohol on sleep-related cardiovascular physiology and long-term health outcomes.

## Figures and Tables

**Figure 1 nutrients-17-01470-f001:**
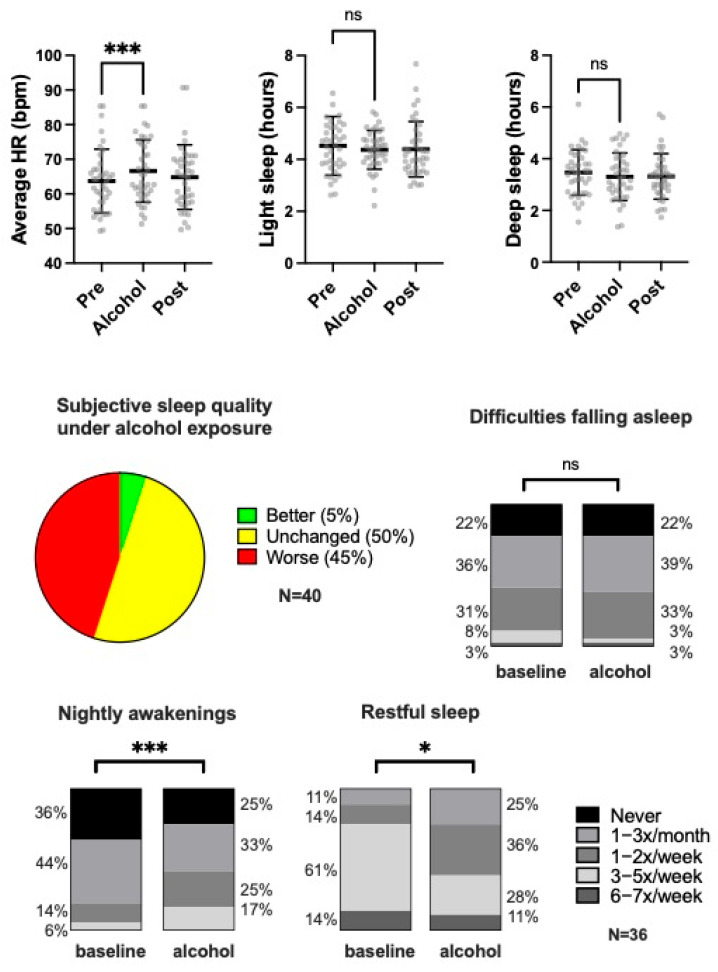
Selected sleep and HR parameters. Depicted are objective results for average HR and light and deep sleep before (pre), during (alcohol), and after (post) exposure to alcohol. Subjective data on sleep quality were assessed as indicated at baseline and after alcohol exposure. Upper panels show mean and SD. Lower panels indicate relative proportions. HR—heart rate; bpm—beats per minute; ns—not significant; ***—*p* < 0.001; *—*p* < 0.05.

**Table 1 nutrients-17-01470-t001:** Results of the PSQI questionnaire at baseline. Data are presented as N(%). Total N = 40. Results of individual components are rated from “++” and “+” to “−” and “−−” in the order of decreasing sleep quality indicators.

PSQI: 6.2 ± 2.5	++	+	−	−−
Quality	5 (13)	26 (65)	9 (23)	0 (0)
Latency	4 (10)	16 (40)	18 (45)	2 (5)
Duration	27 (68)	9 (23)	3 (8)	1 (3)
Efficiency	2 (5)	27 (68)	9 (23)	2 (5)
Disturbance	9 (23)	31 (78)	0 (0)	0 (0)
Medication	0 (0)	2 (5)	2 (5)	1 (3)
Day Dysfunction	9 (23)	15 (38)	15 (38)	1 (3)

**Table 2 nutrients-17-01470-t002:** Smartwatch recordings for sleep, activity, and heart rate parameters. Total N = 40. Data are presented as mean (SD). HR—heart rate; bpm—beats per minute. Comparisons between pre-exposure and alcohol exposure were performed for all variables using one-way ANOVA. Only average HR reached statistical significance, with *p* < 0.05 (*).

	Pre	Alcohol	Post
Light sleep [hours]	4.52 (1.13)	4.37 (0.75)	4.39 (1.07)
Deep sleep [hours]	3.47 (0.88)	3.30 (0.92)	3.31 (0.88)
Time awake [minutes]	22.5 (12.4)	22.9 (11.8)	23.2 (16.1)
Nocturnal awakenings [N]	1.65 (1.14)	1.69 (1.01)	1.66 (1.19)
Sleep latency [minutes]	5.5 (13.2)	2.5 (1.5)	2.5 (1.5)
Wake-up duration [minutes]	4.0 (4.4)	3.3 (3.9)	3.8 (5.3)
Calories (passive) [kcal]	1512 (235)	1522 (240)	1507 (248)
Calories (active) [kcal]	289 (285)	358 (381)	302 (304)
Distance [m]	5827 (2278)	6527 (2940)	6115 (2493)
Steps [N]	7696 (3010)	8413 (3428)	8001 (3066)
HR average [bpm]	63.6 (9.2) *	66.6 (9.0) *	64.9 (9.3)
HR mininum [bpm]	54.6 (8.2)	56.0 (7.8)	55.8 (7.8)
HR maximum [bpm]	82.9 (13.2)	86.1 (10.2)	85.0 (14.8)

## Data Availability

The raw data supporting the conclusions of this article will be made available by the authors on request.

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
