# Peer review of "The Impact of Alcohol on Sleep Physiology: A Prospective Observational Study on Nocturnal Resting Heart Rate Using Smartwatch Technology"

_nutrients, 2025, doi:10.3390/nu17091470_

Round 1
Reviewer 1 Report
Comments and Suggestions for Authors
Introduction
The authors refer to "moderate consumption," but they do not specify what this entails—how many grams of alcohol, what blood alcohol concentration, whether there are gender-specific differences, etc. The mention of "sleep quality" and "sleep architecture" (lines 63–66) is interesting but somewhat vague. It would be helpful to clarify what is meant by sleep quality: fragmentation, REM/NREM duration, or perhaps only subjective perception. Some sentences convey similar information (e.g., that alcohol increases nighttime heart rate), which could be shortened or synthesized for better flow. It would also be useful to end the section with a sentence emphasizing why this study is important.
Materials and Methods
For an observational study, 40 participants is a rather small sample. Allowing participants to choose their alcoholic beverage (beer, wine, etc.) increases ecological validity but reduces control over variables (e.g., differences in metabolism rate, polyphenol content, sugar levels). It is unclear whether the Pittsburgh Sleep Quality Index (PSQI) was used only once (before the study) or also after alcohol exposure—this is important if the authors aim to assess subjective changes.
Results
No significant changes were detected in sleep parameters—this might be due to the small sample size (n=40). Confidence intervals (CIs) for heart rate or sleep outcomes should be reported to improve the clinical interpretability of the findings. Did alcohol affect heart rate differently in physically active individuals? Or smokers? Or between men and women?
Discussion
There is no in-depth analysis of why there were no observed differences in sleep (objectively). It is essential to mention the limitations of smartwatch accuracy compared to polysomnography (e.g., in detecting sleep stages or micro-awakenings). There is no reflection on a possible placebo effect. The subjective worsening of sleep quality may have been partly influenced by participant expectations ("I drank = I sleep worse"). This should be acknowledged as a possible psychological factor.
Conclusions
The phrase "should not be underestimated" is vague. Practical implications are lacking.
Author Response
We sincerely thank the reviewer for the valuable feedback and thoughtful suggestions, which have significantly contributed to improving the clarity, depth, and scientific rigor of our manuscript. Below, we address each comment point by point and indicate the corresponding revisions made in the manuscript.
For detailed revisions: Please see the attachment.

Reviewer 2 Report
Comments and Suggestions for Authors
The paper investigates sleep quality after alcohol consumption, using a smartwatch.
However, to make the publication clearer, it is necessary to:
- add *** to Figure 1 because they are currently not visible, and are listed in the image description
- add similarity or statistical differences labels to Table 2 for the parameters listed in column 1, also the statement "157" in the header of the first column was probably left by mistake
The statements in L127 and L164 are not aligned (8500 or 8000 steps, on average?!)
However, for a scientific paper, I would expect a deeper analysis of the interdependence of sleep quality with anthropometry, gender or similar.
I suggest that the authors add data on the type of alcohol, quantities, and add energy values of consumed drinks, because the calories they list in Table 2 are exclusively related to the information collected from the watch.
Let them show the ranges of quantities with some infographics and definitely emphasize the "French paradox" in the discussion, in which moderate alcohol consumption is the one that has proven to be positive for general health. Perhaps they should also add an analysis of the type of drink by gender. There is data on alcohol consumption in the EU and the world (e.g. https://ec.europa.eu/eurostat/statistics-explained/index.php?title=Alcohol_consumption_statistics) and adding a table of the energy-nutritional content of the average consumption of beer, wine, and spirits will certainly focus the work on the "nutritional side". For example, maybe those who consumed beer, which contains essential nutrients such as vitamins, minerals, and antioxidants, slept better. It is a source of B vitamins, dietary silicon, and some fiber, which can contribute to overall health.
Sincerely
Author Response

(The authors gave the same response as above.)

Reviewer 3 Report
Comments and Suggestions for Authors
Dear corresponding Author, thank you for submitting your work to the jurnal Nutrients and congratulations on your research.
1) Brief Summary Prospective observational study on 40 healthy adults that examined the effect of moderate alcohol consumption on nocturnal heart rate and sleep parameters using smartwatch. Protocol of 9 days (3 baseline, 3 with alcohol, 3 post-exposure). Results show significant increase in nocturnal heart rate during alcohol exposure with rapid normalization post-exposure, without alterations in sleep architecture.
2) General Comments The manuscript addresses a relevant topic with an innovative approach in real-world conditions, however there are some aspects that in my opinion should be improved:
- Limited sample (n=40) and young, healthy population which reduces generalizability and is not properly adressed in the Limitations
- Failure to consider factors such as body weight and habitual alcohol consumption
- Absence of evaluation of chronic effects due to the short duration of the study
- Insufficient discussion on physiological mechanisms underlying the increase in heart rate
- Limited consistency of the device used, which is commercial and has scarce scientific validation
3) Specific Comments
- Lines 81-85: Better specify the methods to verify adherence to the protocol
- Lines 93-96: Justify the choice of alcohol quantities (40g women, 60g men)
- Lines 97-99: Clarify how water intake was monitored on control days T
- able 1: Add PSQI scores for each phase of the study F
- igure 1: Increase text size of the axes and clearly indicate statistical significance
- Lines 115-118: Detail methodology for post-hoc power calculation
- Lines 135-136: Provide quantitative data on the subjective decrease in sleep quality
- Lines 191-192: Elaborate on the clinical significance of the 3 bpm increase compared to the 10 bpm cited in literature
- Include gender-stratified analysis given the different physiological response to alcohol
- Develop the concept of "impairment of physiological recovery" mentioned in the conclusions
In conclusion, the work appears interesting but given the criticisms reported in the general and specific comments, I believe that at the moment it is not suitable for publication but I look forward to reading an improved work
Author Response

(The authors gave the same response as above.)

Round 2
Reviewer 1 Report
Comments and Suggestions for Authors
Accept in present form
Reviewer 2 Report
Comments and Suggestions for Authors
I certainly appreciate the changes and additions to the manuscript.
I only suggest that in the tables that state Pre- & post - (table S1 and table 2) and below it is indicated to whom it refers, as it is stated in Figure 1.
Reviewer 3 Report
Comments and Suggestions for Authors
I have carefully reviewed the authors' revisions and I appreciate the significant contribution and the effort they made to improve this version of the manuscript. I believe that, in its current form, the study can be recommended for publication.